# Aggressiveness and Patulin Production in *Penicillium expansum* Multidrug Resistant Strains with Different Expression Levels of MFS and ABC Transporters, in the Presence or Absence of Fludioxonil

**DOI:** 10.3390/plants12061398

**Published:** 2023-03-21

**Authors:** Panagiota Ntasiou, Anastasios Samaras, Emmanouil-Nikolaos Papadakis, Urania Menkissoglu-Spiroudi, George S. Karaoglanidis

**Affiliations:** 1Laboratory of Plant Pathology, School of Agriculture, Faculty of Agriculture, Forestry and Natural Environment, Aristotle University of Thessaloniki, 54124 Thessaloniki, Greece; ntasioup@agro.auth.gr (P.N.); samarasanast@gmail.com (A.S.); 2Pesticide Science Laboratory, School of Agriculture, Faculty of Agriculture, Forestry and Natural Environment, Aristotle University of Thessaloniki, 54124 Thessaloniki, Greece; papadakm@agro.auth.gr (E.-N.P.); rmenkis@agro.auth.gr (U.M.-S.)

**Keywords:** blue mold disease, patulin analysis, MFS-transporters, ABC-transporters, apple fruit, fludioxonil

## Abstract

*Penicillium expansum* is the most common postharvest pathogen of apple fruit, causing blue mold disease. Due to the extensive use of fungicides, strains resistant to multiple chemical classes have been selected. A previous study by our group proposed that the overexpression of MFS (major facilitator superfamily) and ABC (ATP binding cassette) transporters constitute an alternative resistance mechanism in Multi Drug resistant (MDR) strains of this pathogen. This study was initiated to determine two main biological fitness parameters of MDR strains: aggressiveness against apple fruit and patulin production. In addition, the expression pattern of efflux transporters and hydroxylase-encoding genes that belong to the patulin biosynthesis pathway, in the presence or absence of fludioxonil and under in vitro and in vivo conditions were investigated. Results showed that the MDR strains produced higher concentrations of patulin but showed a lower pathogenicity compared to the wild-type isolates. Moreover, expression analysis of *patC*, *patM* and *patH* genes indicated that the higher expression levels do not correlate with the detected patulin concentration. The selection of MDR strains in *P. expansum* populations and the fact that they produce more patulin, constitutes a serious concern not only for successful disease control but also for human health. The above-mentioned data represent the first report of MDR in *P. expansum* associated with its patulin-production ability and the expression level of patulin biosynthesis pathway genes.

## 1. Introduction

Apples are one of the most important crops in Greece, based on their economic impact. Apple fruit is characterized by its long storage ability, and this has a significant impact on the price, quality, and availability of the product. However, longer storage time gives the opportunity for the development of postharvest pathogens. *Penicillium expansum* is the most common postharvest pathogen of apple fruit, causing soft rot of the fruit, known as blue mold. In addition to quantitative yield loss, it may reduce the quality of the fruit and its by-products by the production of patulin, a mycotoxin that causes gastrointestinal disorders and chronic problems, including genotoxicity, immunotoxicity and neurotoxicity [1,2]. Patulin, 2-Hydroxy-3,7-dioxabicyclo[4.3.0]nona-5,9-dien-8-one, is an organic compound with a lactonic structure, soluble in water and stable in acidic solution and heat, which is therefore not destroyed during thermal degradation or pasteurization [3,4]. Several factors and their interactions such as apple variety, fungal strain, and the physiological properties of the fruit affect patulin production [5].

The patulin biosynthesis pathway consists of a cascade of enzymatic action, whose first committed step is the condensation of one acetyl CoA and 3 units of malonyl CoA [6]. Patulin production is controlled by a 15-gene cluster, which was identified as *P. expansum* [7]. Previous studies described the patulin cluster in detail, which has a size of 44-kb (*PatA* to *PatO*). These genes encode the enzymes necessary for the biosynthesis of the mycotoxin, as well as the specific regulatory factors and transporters. Among these 15 genes, 11 encode biosynthetic enzymes, three encode transporters (*PatA, PatC*, and *PatM*), and one encodes a putative transcription factor (*PatL*). The genes *PatA, PatC*, and *PatM* encode an acetate transporter, an MFS transporter, and an ABC transporter, respectively [8]. *PatH* encodes the enzyme cytochromes P450 (responsible for the hydroxylation of m-cresol to m-hydroxybenzyl alcohol) [1,9]. Interestingly, in patulin permissive conditions, an over-expression pattern was observed in all 15 genes [7,10,11]. Numerous studies have investigated the role of patulin in infection and pathogenicity of *P. expansum* but with controversial findings. On one hand, a previous report suggested that the disruption of the 6-methyl-salicylic acid synthase gene sequence resulted in reduced virulence of the disrupted mutants [12]. On the other hand, more recent reports suggested that patulin-related gene deletion did not affect the pathogenic ability of the deletion mutants and that patulin was not required by *P. expansum* to infect apple fruits but that it can act as a cultivar-dependent aggressiveness factor [13,14].

Chemical control is the main method used to combat blue mold in apple fruit. Fungicides are applied as either preharvest treatments in the orchard a few days before harvest, as postharvest treatments applied by fruit drenching before storage, or as on-line spray application in storage facilities. Fungicides from different chemical classes are used, such as anilinopyrimidines (cyprodinil, pyrimethanil), phenylpyrroles (fludioxonil), quinone outside inhibitors (QoIs) (pyraclostrobin, trifloxystrobin), and the succinate dehydrogenase inhibitors (SDHIs) (boscalid, penthiopyrad, isopyrazam). However, development of fungicide resistance represents one of the major limitations of chemical control. In the past, the development of resistance of *P. expansum* to benzimidazoles or DMI fungicides associated with target-site modifications have been reported [15,16]. Recent studies conducted in our laboratory have shown that in Greece, *P. expansum* isolates with simultaneous resistance to different active ingredients (so-called Multi Drug Resistance, MDR) have been selected within the fungal population [17]. Whole transcriptome analysis of MDR and sensitive isolates in the presence or absence of fludioxonil revealed that the MDR phenotype was associated with the overexpression of MFS and ABC transporters [17]. More specifically, *PeABC1*, *PeMFS1* and *PeMFS2* genes showed higher expression even in the absence of the fungicide treatment. In addition, 18 ABC and MFS transporters induced 4 hpi in resistant strains with the fludioxonil treatment [17]. Previously, overexpression of ABC or MFS transporters has been described as the mechanism of multidrug resistance development in *Botrytis cinerea* [18,19], *Zymoseptrotia tritici* [20], *Oculimacula yallundae* [21], *Sclerotinia homoecarpa* [22] and the closely related to *P. expansum* agent of citrus Green Mold disease *P. digitatum* [23,24,25,26,27,28].

In the past, numerous studies investigated the role of ABC and MFS transporters in the secretion of endogenous or exogenous toxins. Starting from 1994, Bissinging and Kuchler [29] reported that PDR5, a well-known ABC transporter provided protection against a mycotoxin produced from *Phylomyces chartarum*. Other studies reported that efflux pumps were involved in the secretion of toxins, such as cercosporin in *Cercospora kikuchii* and *Cercospora nicotianae*, in combination with the MFS transporters CFP and CTB4, and the ABC transporter ATR1 [30,31,32]. Additionally, two mycotoxin transporters, Tri12 and ZRA1, have been described in *Fusarium graminearum*. Tri12 is an MFS transporter that allows trichothecene secretion [33] and ZRA1 is an ABC transporter responsible for zearalenone transport [34]. Another study indicates that the MFS gene *afl*T on *Aspergillus flavus* belongs to the aflatoxin biosynthesis pathway, but does not play a significant role in aflatoxin secretion [35]. The information available indicates that secretion of endogenously-produced mycotoxins could proceed via ABC and/or MFS transporters.

The recent detection of *P. expansum* strains with multidrug resistance phenotype to fungicides due to overexpression of ABC and MFS transporters [17] gives rise to questions related to the risk for further selection of these strains that may lead to insufficient disease control and to the risk for increased patulin contamination in apple byproducts derived from fruit infected by these MDR strains. Therefore, this study was initiated, aiming to compare aspects of the aggressiveness and patulin production of MDR and sensitive strains of *P. expansum* in the presence or absence of fungicide. In addition, a gene expression analysis was conducted to investigate how the exposure to fungicides affects patulin production and, in particular, whether genes encoding transporters in the patulin biosynthesis pathway have a role in the production or transport/secretion of patulin.

## 2. Results

### 2.1. Aggressiveness Measurements

To determine the aggressiveness of the MDR strains in comparison to those of the wild-type isolates, 14 MDR and four sensitive isolates were used, to artificially inoculate apple fruit, both treated and untreated with fludioxonil. The application of fludioxonil significantly reduced disease severity in fruit inoculated with the sensitive stains, while on fruit inoculated by the MDR strains disease severity remained almost unaffected (Figure 1). However, the aggressiveness of the MDR isolates on the untreated fruit was significantly lower (*p* < 0.05) compared to that of the sensitive isolates. Lesion diameters caused by MDR isolates had values ranging from 2 to 3 cm, while for the sensitive isolates the respective values were 3.4 to 3.5 cm (Table 1). As expected, the application of fludioxonil onto the apple fruit failed to provide high control efficacy when the fruits were artificially inoculated with the MDR isolates, with values ranging from 16.67 to 33.33% (Table 1). On the other hand, control efficacy achieved by fludioxonil treatments was higher for the sensitive isolates, with values ranging from 51.43 to 57.14%.

### 2.2. Validation of Patulin Determination Analytical Method

Recoveries for patulin ranged from 72 to 110% οn PDA and from 84 to 114% in apple samples and the respective relative standard deviation values (RSDs %) of all % mean recovery values were <8% and <13% in PDA and apple samples, respectively. The calibration curves were linear with correlation coefficients (R^2^) values > 0.99 in the concentration range studied.

### 2.3. Patulin Production

All the isolates but one MDR isolate included in the study were found to be able to produce patulin both in vitro and in vivo (Appendix A). Patulin was produced in vitro by *P. expansum* MDR and sensitive isolates at a range of 252.7 to 826 and 210.7 to 409.8 μg/g, respectively (Table 2). As a mean, MDR isolates produced significantly higher (*p* < 0.05) concentrations of patulin compared to sensitive isolates in the absence of fungicide. However, no difference (*p* > 0.05) was observed in mean patulin concentration produced by the MDR and the sensitive isolates in the presence of fludioxonil in vitro, as the mean values were 371.4 and 327.3 μg/g, respectively (Table 2).

The analysis conducted on apple fruit showed that patulin was produced in lower concentrations. Patulin was produced by all MDR and sensitive isolates at concentrations ranging from 0.2 to 32.5 and 0.34 to 13.35 μg/g, respectively (Table 2). No difference (*p* > 0.05) was observed in mean patulin concentration produced by the MDR and sensitive isolates in the absence of fludioxonil in vivo, as the mean values were 11.4 and 5.20 μg/g, respectively. Patulin production was higher (*p* < 0.05) on fludioxonil-treated fruit both for MDR and the sensitive isolates with mean values of 25.5 and 16.7 μg/g, respectively.

### 2.4. Expression Levels of Patulin Biosynthesis Genes on Fludioxonil-Treated Apple Fruit

Besides the measurements of patulin production on apple fruit, the expression levels of *patC, patM and patH* genes on fludioxonil-treated fruit were determined, 48 and 72 h after inoculation. Resistant isolates, such as HGS15, HF2, ARD11, HG4 and AGS5, also showed high patulin production and significant upregulation pattern in all the three examined genes (Figure 2). Furthermore, variable expression patterns were observed with respect to the sampling time. For instance, the *patC* gene was found to be significantly up-regulated at 72 hpi in HGS15 isolate, while in isolates HF2 and HG4 the same gene was found to be upregulated at 96 hpi. Similarly, the *patH* gene was found to be upregulated in isolates HGS16 and AGS5 at 72 hpi, while up-regulation of the same gene in isolates HF2, HG4 and ARD11 was observed at 96 hpi (Figure 2).

### 2.5. Gene Regulation of Patulin Biosynthesis Genes in MDR Isolates Grown In Vitro in the Presence of Fludioxonil

To investigate the expression pattern of the patulin biosynthesis pathway genes (*patC, patH, patM*) under in vitro conditions, we proceeded to a qPCR analysis in MDR and sensitive isolates grown on PDA dishes in the presence of fludioxonil. Our analysis showed that under in vitro conditions, a higher expression of all three genes was observed at 72 hpi in most of the isolates. More specifically, the *patC* gene was found to be upregulated significantly in HGS15 and HF2 isolates, *patM* was overexpressed in HGS15 and ARD11, while *patH* showed higher expression only in ARD11 isolate (Figure 3).

The comparison of expression levels between selected resistant isolates, normalized with the expression level of sensitive isolates, showed that *patC* genes were overexpressed significantly with isolate HG4. Expression patterns of *patM* gene showed low diversity, and only isolate ARD11 had significantly higher expression. At the same time point (48 hpi), the expression of the *patH* gene showed a higher variability among the isolates. More specifically, AGS5 and HF2 isolates had low expression levels, followed by HG4, while the higher expression levels were observed in HGS15 and ARD11 isolates (Figure 4).

## 3. Discussion

*P. expansum* is considered as a high-risk pathogen for the development of resistance to fungicides [36]. The pathogen is exposed to high fungicide selection pressure as a target of postharvest fungicide treatments applied in packinghouses as well as a target or non-target pathogen of applications conducted in the orchards before harvest. Recently in Greece, the presence of strains with a multidrug resistance phenotype associated with overexpression of efflux transporters was reported [17]. The presence of MDR strains within the fungal population may represent a major threat for successful control of the pathogen. In this study, we proceeded with the characterization of MDR strains in terms of aggressiveness against apple fruit and patulin production capacity determination as a tool to predict the risk for MDR phenotype increase in frequency within the fungal population or the risk for increased patulin concentrations in the apple byproducts. The development and evolution of fungicide resistance in fungal populations is largely dependent on the dynamics of competition between fungicide resistant and sensitive strains, which are affected by the fitness of the strains [37]. Fungicide resistance would be limited if a resistant subpopulation had lower parasitic or saprophytic fitness. In contrast, absence of fitness costs in the resistant fraction of the population would lead to a stable resistance frequency in the absence of fungicide selection force, or to rapid development and evolution of resistance under the fungicide selection force [38]. The measurements were conducted both in the absence and the presence of the fungicide fludioxonil.

Fungicide treatments may select for MDR strains in the field based on the selective advantage conferred by the MDR phenotype in fungal strains as has been reported previously in the case of *B. cinerea* [18]. However, in the case of *P. expansum*, it is of crucial importance whether the MDR strains can survive in the field or the packinghouses in the absence of fungicide selection pressure. The results of our study revealed that MDR strains showed a reduced pathogenicity compared to that of the wild type ones. Previous studies showed that efflux pump function in microorganisms is energy-dependent. Therefore, it is expected that overexpression of efflux pumps may have a cost, leading to reduced fitness. In the past, decreased fitness has been observed not only in MDR strains of plant pathogenic fungi but also in bacterial species with multi-antibiotic resistance [39,40].

The effects of fungicides applied against fungal pathogens on mycotoxin production is a major food safety concern highlighted by the European Commission Scientific Committee in Plants [41]. In this frame, a maximum limit of 50 μg/kg on fruit juices and drinks, and a limit of 25 μg/kg on solid apple products, has been set by European Union (EU), while an even lower limit of 10 μg/kg is accepted for certain infants’ foods [42]. Agricultural products may show an increased contamination by mycotoxins because of insufficient control of mycotoxigenic fungi due to fungicide resistance development. Furthermore, fungicide resistance development may affect the mycotoxigenic ability of the resistant isolates and may lead to increased accumulation of mycotoxins in the infected products. Previous studies have shown that resistance to the DMI fungicide difenoconazole was associated with an increase of the trichothecene mycotoxin 3-acetyl deoxynivalenol (3- ADON) in *Fusarium culmorum* [43]. Similarly, *F. sporotrichioides* strains with resistance to carbendazim produced the mycotoxins T-2 toxin, 4,15 diacetoxyscirpenol, and neosolaniol in higher concentrations as opposed to sensitive strains [44] and benzimidazole resistance increased the trichothecene production of *F. graminearum* [45]. Similarly, in the use of mutant strains of *Aspergillus parasiticus, A. carbonarius* and *P. expansum* with laboratory-induced resistance to fludioxonil, resistant isolates were found to produce significantly higher levels of mycotoxins (aflatoxins, ochratoxins and patulin) compared to the sensitive strains [46,47,48,49]. Patulin production in most *P. expansum* boscalid, benzimidazole and cyprodinil-resistant strains was significantly elevated compared the wild-type parent strain both in vitro and in vivo [15,38,50]. In contrast, in some other cases, development of resistance to fludioxonil in *F. asiaticum*, was associated with reduced production of 3ADON, D3G and DON [51]. In our study, patulin production in most MDR isolates was significantly higher compared to isolates of wild type, both on artificial nutrient media (in vitro) and on apple tissue (in vivo). Interestingly, both the MDR and the sensitive isolates produced higher amounts of patulin in the presence of fludioxonil either in vitro or on apple fruit. The impact of the exposure to fungicide on patulin production was stronger on apple fruit for both MDR and sensitive isolates. Thus, exposure of MDR isolates to fungicides may not only lead to insufficient control as our results suggested, but in addition, may lead to increased mycotoxin accumulation. The results of our study are in agreement with previous findings suggesting that exposure to several fungicides had stimulated DON biosynthesis in vitro and in vivo by *F. graminearum* [52,53,54]. Furthermore, our results are in agreement with findings reported by Paterson [55] suggesting that patulin production was enhanced in *P. expansum* strains grown on nutrient media amended with the fungicide captan. The quantity of patulin correlated with the reduction in colony diameter caused by the fungicide and the patulin production was higher in fungicide-treated colonies [55]. In addition to these data derived from in vitro measurements, Morales et al. [56] analyzed patulin accumulation in fungicide-treated apples. The lesions caused by *P. expansum* were smaller in treated apples, but patulin accumulation was not significantly differentiated. The accumulation of patulin was expressed in ng of patulin per apple, which may give an idea of the total production of patulin regardless of the mycelial mass. Thus, it seems that some fungicides may enhance the secondary metabolism, thus increasing the quantity of patulin per weight of rotten tissue. Considering that, if the fungicide is efficient in reducing fungal growth, total patulin accumulation per apple should not be higher. It may be assumed that in the presence of sub-lethal concentrations of certain fungicides the fungal strains respond to this stress by increasing production of secondary metabolites including mycotoxins [57].

To investigate whether there was any relation between the patulin concentration and the induction of genes that belong to the patulin biosynthesis pathway, we proceeded with an expression analysis of some of these genes. The selected genes encoded one ABC (*patC*), one MFS (*patM*) transporter, and one P450 hydroxylase (*patH*). ABC and MFS transporters have a remarkable general substrate specificity, and in addition to synthetic toxic compounds, they are able to transport a wide variety of either endogenous or exogenous toxins [58]. The selected genes in MDR strains grown on fludioxonil-treated apple fruit showed a variability in expression patterns, in samples collected 72 and 96 h after infection. However, the findings of our study suggested that there was no direct correlation between the measured patulin concentration and the expression levels of the three tested genes. It has been proposed that the patulin biosynthesis pathway is controlled by a 15-gene cluster. However, all the genes of the cluster have not shown the same influence on patulin production and the detectable patulin cannot be related to the gene expression of these genes individually [10]. In the same study, the *patA* gene, which encodes a putative acetate transporter, showed an excessive expression under patulin restrictive conditions, suggesting that *patA* gene transcription is regulated differently to transcription of the other cluster genes [10]. Moreover, previous studies have shown that additional genes (*PepatE, PepatN, PepatK, PepatG*, and *PepatH*) play an important role in patulin biosynthesis. Their up-regulation at a favorable pH (pH 5.0) correlates directly with patulin production [7]. However, the expression levels of *patH* in the tested isolates in our study did not follow the same pattern. Such a contrast might be related to either a data inadequacy of significant gene expression in patulin biosynthesis pathway or the impact of fludioxonil treatment on apple fruits. Furthermore, expression of genes participating in secondary metabolism, such as patulin, is usually strictly controlled by environmental factors, and regulated at transcriptional level [59].

One more conclusion that we can generate based on the findings of our work is that the ABC and MFS transporters associated with the MDR phenotype in *P. expansum* isolates used in this study are not involved in patulin secretion. Based on data that we generated in a previous publication with the same set of isolates [17] and from the gene expression analysis of the patulin biosynthesis genes, we consider that the ABC and MFS transporters from *P. expansum* do not play role in patulin secretion from fungal cells. In contrast to our findings, other studies reported that some transporters were involved in the secretion of endogenous or exogenous toxins [32,35,60]. For instance, FsTri12p is an MFS transporter that plays a role in self-protection toward trichothecenes for both *F. sporotrichioides* and *F. graminearum*. Tri12 mutants of both species grow more slowly than the wild type strains under trichothecene biosynthesis-inducing conditions but not under trichothecene-restrictive conditions [33]. On the other hand, another study found that an MFS transporter of *Aspergillus parasiticus* (*aflT*) that participates in the aflatoxin biosynthesis pathway is not involved in aflatoxin secretion [35]. The involvement of *patC*, *path,* and *patM* genes in patulin biosynthesis should be clarified with the generation of knockout mutants’ strains and investigation of how the deletion of these genes affects patulin production, virulence against apple fruits and resistant phenotype.

The expression analysis of genes belonging to the cluster of patulin biosynthesis genes showed that there was no correlation between induction of those genes and virulence of the tested strains. Such a finding is in agreement with the findings of a previous study suggesting that the deletion of some of the other genes of the patulin cluster (*patK* and *patL*) did not affect the virulence of *P. expansum* [7]. This conclusion is in line with the fact that most of the fungal species that are able to produce patulin are not plant pathogens.

## 4. Materials and Methods

### 4.1. Pathogen Isolates

Fourteen (14) isolates of *P. expansum* exhibiting a multidrug resistance phenotype (MDR) to several fungicides and four fungicide-sensitive fungal isolates were used in the study. The isolates had been collected for the requirements of a recently published report aiming to investigate the resistance mechanism in MDR isolates [17]. In the above mentioned study, the MDR phenotype of the resistant isolates was associated with overexpression of MFS and ABC transporter genes at variable levels. The isolates used in this study were maintained at 4 °C on petri dishes containing potato dextrose agar (PDA) until further use.

### 4.2. Aggressiveness Measurements on Apple Fruit

The aggressiveness of *P. expansum* MDR and sensitive isolates was determined by measuring the lesion diameter caused by each isolate on fludioxonil-treated or untreated apple fruit (cv. Red Delicious). Prior to inoculation, fruits were surface-disinfected for 5 min by drenching them in a 1% sodium hypochlorite solution, rinsed three times with sterile-deionized water and air-dried. Following disinfection, the fruits were subsequently immersed for 5 min by soaking them in a 5 μg mL^−1^ fludioxonil aqueous solution and dried for 1 h at room temperature. The fruit were wound-inoculated at the equator. Three wounds per fruit were made using a 2-mm diameter nail head into 3 mm in depth and on each wound 20 μL of a conidial suspension containing 1 × 10^5^ conidia per ml was placed with a pipette. In total, 15 fludioxonil-treated and 15 untreated fruits per isolate were inoculated and the experiment was replicated twice. Non-inoculated fruits treated with sterile water were used as control. The fruits were placed on wire mesh platforms (10 fruit per box) in plastic boxes (23 × 31 × 10 cm [length × width × height]). Twenty ml of water was added in each box and then the boxes were covered to maintain high relative humidity. The inoculated fruits were incubated for 7 days at 25 °C and the aggressiveness of the isolates was scored by measuring the lesion diameter.

### 4.3. Reagents and Chemicals for Mycotoxin Detection and Quantification

Acetonitrile, water, formic acid, and ethyl acetate, methanol (all of HPLC grade) were obtained from Merck (Darmstadt, Germany) and Chem-Lab (Zedelgem, Belgium), respectively. Analytical-grade sodium chloride and magnesium sulfate anhydrous (fine) were from Sigma-Aldrich (St Luis, MO, USA). Anhydrous magnesium sulfate (coarse) of technical grade was obtained from Panreac (Barcelona, Spain), while primary-secondary amine (PSA) sorbent was purchased from Agilent (Santa Clara, CA, USA). Certified standard of patulin (PAT, 99,7%) was obtained from Romer Labs Biopure (Romer Labs, Tulln, Austria).

### 4.4. Patulin Extraction Procedure from Artificial Nutrient Media and Apple Fruit

For the determination of patulin production in vitro the isolates were cultured in 9 cm Petri dishes (three replicate plates per isolate) containing Potato Dextrose Agar (PDA) media in the absence and presence of fludioxonil [61]. The concentrations of fludioxonil in in vitro treatments were 0.05 and 0.6 μg mL^−1^ for sensitive and MDR strains, respectively. Petri dishes were incubated at 25 °C, in the dark, for 7 days. The extraction of patulin was performed using the method described by Andersen et al. [62], slightly modified. After incubation, agar plugs (0.5 g) were cut form the Petri dish and transferred into 5 mL screw-cap vials. The plugs were extracted with 2 mL of ethyl acetate containing 1% formic acid by sonication for 60 min. The extracts were transferred into 10 mL glass tubes and concentrated to dryness with the use of nitrogen stream at 30 °C. The extracts were re-dissolved ultrasonically in 2 mL of ethyl acetate, filtered through 0.45 μm polytetrafluoroethylene (PTFE)syringe filter and transferred into autosampler vials for instrumental analysis by HPLC.

For the determination of patulin production in vivo, the apples (cv. Red Delicious) were artificially inoculated as described previously. In total, five fruits were inoculated per isolate and the fruits were incubated at 25 °C for 7 days. The extraction of patulin from the apples was performed according to the QuEChERS (Quick, Easy, Cheap, Efficient, Rugged, and Safe) method [63], which was slightly modified and optimized to better suit the extraction of patulin from the apples. In brief, 2.5 gr of pulped sample was placed into a 50-mL polypropylene screw-cap falcon tube, and 2 mL of ultra-pure water was added. The contents were briefly mixed and 5 mL of 1% acetic acid in acetonitrile were added. The sample was vortexed for 1 min and then 2 g of anhydrous MgSO_4_ (coarse) and 0.5 g of NaCl were added. The sample was vortexed immediately for 1 min. The sample was centrifuged at 5000 rpm for 5 min and 2 mL of the clear supernatant was transferred into a 15 mL screw-cap falcon tube, which contained a sorbent mixture consisting of 50 mg PSA, and 300 mg anhydrous MgSO_4_ (fine). The sample was vortexed for 1 min and then centrifuged at 4000 rpm for 5 min. Afterwards, 1.4 mL of the supernatant was transferred into a glass test tube, containing 20 μL of 2.5% formic acid in acetonitrile. The sample was evaporated to dryness under a gentle nitrogen stream and re-dissolved into 70 μL methanol and 630 μL of mobile phase (containing 95% water with 0.1 % acetic acid and 5% acetonitrile with 0.1% acetic acid). The reconstituted solution was transferred into an injection vial to be analyzed by HPLC.

### 4.5. Instrumental Analysis

Analyses of mycotoxins were carried out using a SpectraSYSTEM high performance liquid chromatograph (Thermo Separation Products, Austin, TX, USA). The HPLC system consisted of a P4000 tertiary solvent pump, a vacuum degasser TSP, an AS3000 autosampler equipped with a 100-μL injection loop, and a UV6000LP diode array detector. Chromatographic separation was done on a Hypersil BDS-C18 (Thermo Finnigan, San Jose, California, USA) column (250 × 4.6 mm, 5 μm) with a 10 × 4 mm (inner diameter) Hypersil BDS-C18 guard column (Thermo Finnigan, USA).

For the analysis of the agar plug extracts, a mixture of 90:10 (% *v*/*v*) water: acetonitrile was used as the mobile phase. Total analysis time was 15 min. The flowrate was 0.7 mL/min and the total run time was 15 min. Injection volume was set at 20 μL. For the analysis of the apple samples, gradient elution was employed. Mobile phase A consisted of 0.1% acetic acid in water and mobile phase B of 0.1% acetic acid in acetonitrile. For the analysis a linear gradient was employed, as shown in Appendix A. The flowrate was 1 mL/min and the total analysis time was 27.5 min. Injection volume was set at 50 μL. Column temperature was set at 30 °C in both methods. Patulin detection was carried out at 275 nm (λ_max_).

### 4.6. Method Validation

The analytical methods were validated for trueness, precision, and linearity by analyzing uncontaminated blank samples (agar and apple tissue), with no presence of patulin, fortified with known amounts of working standard solutions. This procedure was done at three different concentration levels (0.1, 0.5, 1 μg/g PDA and 0.1, 1, 10 μg/g apple tissue). Fortified samples were extracted and analyzed using the proposed method. Five replicates were prepared for each concentration level.

### 4.7. Induction of Patulin Biosynthesis Related Genes In Vitro and In Vivo Samples with qRT-PCR

To determine the transcript changes that took place in patulin biosynthesis-related genes in MDR strains after treatment with fludioxonil, an experiment with in vitro and in vivo treatments was designed. For this purpose, three patulin biosynthesis-related genes (*patC*, *patM,* and *patH*) encoding a MFS transporter, an ABC transporter, and a cytochrome P450, respectively, were selected for expression analysis.

Measurements of gene expression was conducted both in vitro and in vivo on apple fruit during the course of infection. Samples for RNA extraction were collected 0, 72, and 96 h post-inoculation with the pathogen from the apple fruits and 0, 48, and 72 h post-inoculation with the pathogen from the PDA plates. The time points for the transcription analysis were selected according to the time required for pathogen infection to increase the probability of detecting transcript changes [61]. Four biological replicates were used for each time point. Each sample was pulled from three fruits. Samples were put on liquid nitrogen and kept at −80 °C until RNA extraction.

The collected samples were soaked in liquid nitrogen. Total RNA from in vitro samples was extracted using Monarch Total RNA Miniprep Kit (NEB #T2010) according to the manufacturer‘s instructions (NEB, UK). Additionally, total RNA from apple tissues was extracted following the method described by Johnson and Zhu, (2015). The qRT-PCR reactions were carried out on a StepOne Plus Real-Time PCR System (Applied Biosystems, USA) using a SYBR green-based kit (Luna^®^ Universal One-Step RT-qPCR Kit, NEB, UK) according to the manufacturer’s instructions. Amplification conditions were 55 °C for 10 min, 95 °C for 2 min, followed by 35 cycles of 95 °C for 5 s, and 60 °C for 30 s. In all the reactions, samples were run in triplicate. The threshold cycle (CT) was determined using the default threshold settings. The 2^−∆∆Ct^ method was employed to calculate the relative gene expression levels [64]. β-tubulin gene was used as endogenous control. Primers for this analysis are listed in Appendix A.

### 4.8. Statistical Analysis

Data of the two independent replications for aggressiveness test and mean patulin production in vitro and in vivo were combined after testing homogeneity of variance using the Levene test. Mean values among treatments were compared using the pairwise Student’s *t*-test at *p* = 0.05.

The β-tubulin gene was used to normalize the gene expression data and the 2^−∆∆Ct^ method was used to calculate the relative transcripts. Statistical analysis was conducted using Tukey’s test. All statistical analyses were supported by SPSS 21.0 (SPSS, Chicago, IL, USA).

## 5. Conclusions

In this study we determined the pathogenicity and the patulin production ability of *P. expansum* isolates exhibiting an MDR phenotype associated with overexpression of ABC and MFS transporters, in the presence or absence of fludioxonil. Τhe MDR isolates showed a higher patulin production ability compared to that of wild-type isolates both in the presence and the absence of fludioxonil. However, the increased patulin production in MDR strains was not associated with higher aggressiveness on apple fruit. In contrast, wild-type isolates were more aggressive on untreated fruit. Expression data of three genes that participate in the patulin biosynthesis cluster showed that there was no direct correlation between the resistant isolates and the expression pattern of the three tested genes. Results showed that the resistance strains had an up-regulated tense, but this couldn’t connect with patulin production and pathogenicity ability. Such findings provide support the theory suggesting that the expression levels of patulin biosynthesis pathway genes are not directly related to the amount of produced patulin.

## Figures and Tables

**Figure 1 plants-12-01398-f001:**
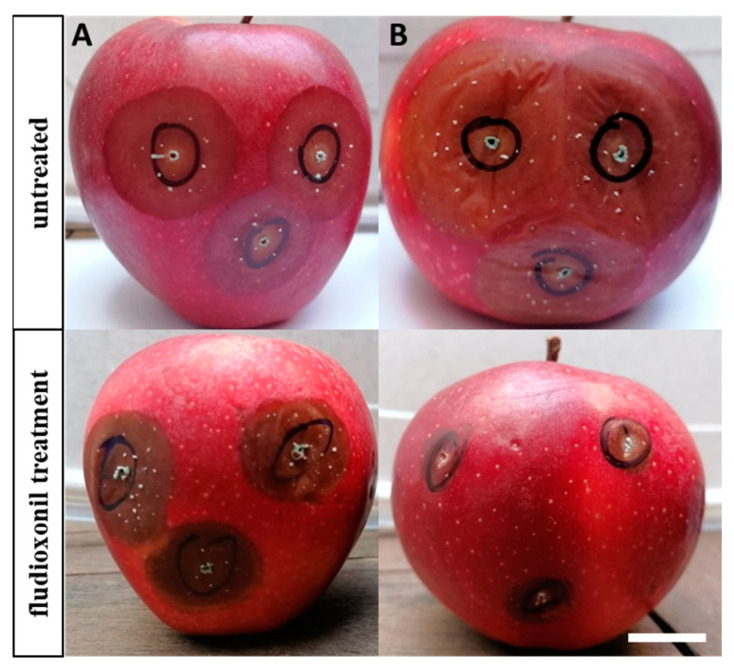
Apple fruit (cv. Red Delicious) artificially inoculated with (**A**) a fungicide-resistant; and (**B**) a sensitive strain of *P. expansum* in the absence (up) and in the presence (down) of fludioxonil (scale bar: 1 cm).

**Figure 2 plants-12-01398-f002:**
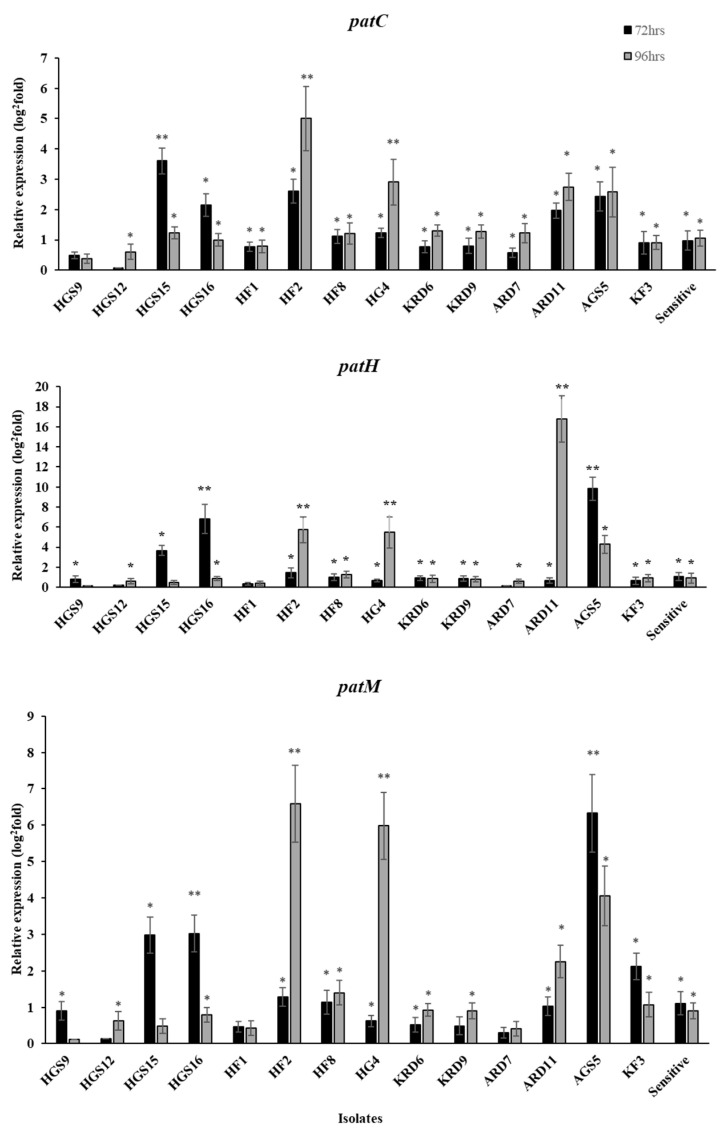
Expression levels of patulin biosynthesis pathway genes (*patC, patH, patM*) in *Penicillium expansum* isolates used to inoculate fludioxonil-treated apple fruit (cv. Red Delicious). Expression levels were analyzed by qRT-PCR at 72 and 96 h post inoculation, and the cDNA samples were standardized using the fludioxonil-untreated fruits at the same time points as references and normalized using the endogenous *b-tubulin* gene. Asterisks indicate the significant differences between treatments (* means differences among the reference and ** means differences between time points) according to Tukey’s test (*p* < 0.05). Vertical lines on the columns indicate the standard error of the mean.

**Figure 3 plants-12-01398-f003:**
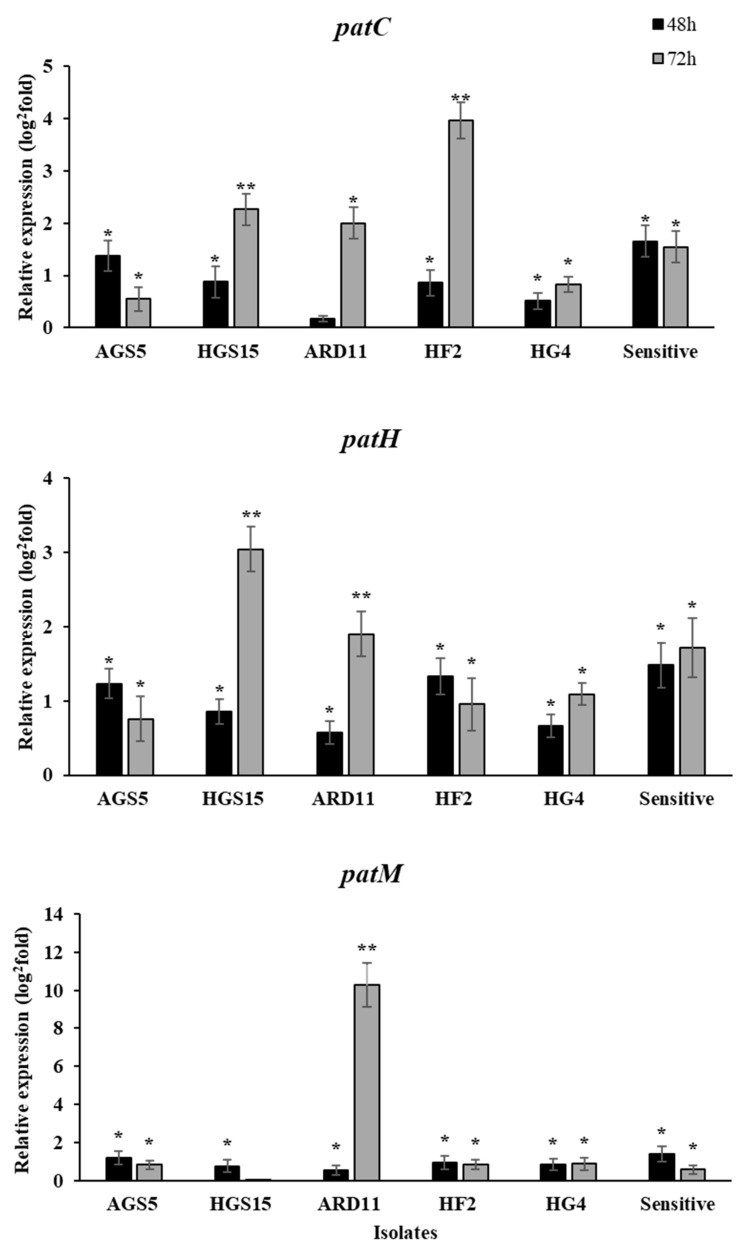
Expression levels of patulin biosynthesis pathway genes (*patC, patH, patM*) in *Penicillium expansum* isolates, grown on PDA media in the presence of fludioxonil. Expression levels were analyzed by qRT-PCR at 48 and 72 h, and the cDNA samples were standardized using mock PDA dishes at the same time points as references and normalized using the endogenous *b-tubulin* gene. Asterisks indicate significant differences between treatments (* means differences among the reference and ** means differences between time points) according to Tukey’s test (*p* < 0.05). Vertical lines on the columns indicate the standard error of the mean.

**Figure 4 plants-12-01398-f004:**
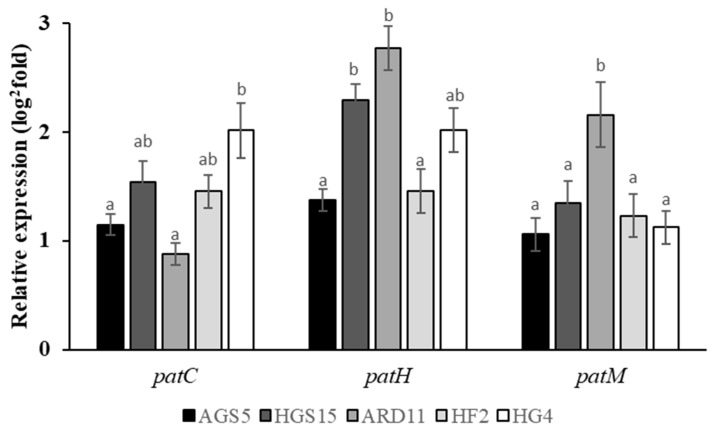
Expression levels of patulin biosynthesis pathway genes (*patC, patH, patM*) *in P. expansum* isolates, grown on PDA medium. Expression levels were analyzed by qRT-PCR at 48 h and the cDNA samples were standardized using the sensitive isolate as reference and were normalized using the endogenous *b-tubulin* gene. Different letters on the columns indicate statistically significant differences between the isolates according to Tukey’s test (*p* < 0.05). Vertical lines on the columns indicate the standard error of the mean.

**Table 1 plants-12-01398-t001:** Aggressiveness, in terms of lesion diameter (cm), of *Penicillium expansum* MDR and sensitive isolates on fludioxonil-treated or untreated apple fruit (cv. Red Delicious), 7 days post inoculation.

Isolate	Phenotype	Lesion Diameter (cm)		Control Efficacy %
		Untreated Fruit	Fludioxonil-Treated Fruit	
HGS9	MDR	2	1.6	20
HF2		2	1.6	20
HF8		3	2	33.3
HF1		2.8	2	28.5
KRD6		3	2	33.3
KRD9		2	1.5	25
ARD7		2.8	2	28.5
ARD11		3	2.5	16.6
HG4		2.7	2	25.9
HGS12		3	2.4	20
HGS15		3	2.5	16.6
HGS16		3	2.5	16.6
KF3		3	2.3	23.3
AGS5		3	2.3	23.3
**Group mean**		**2.7a ****	**2.1a**	**23.67a**
ZRD13	Sensitive	3.4	1.5	55.8
KF14		3.4	1.5	55.8
HF5		3.5	1.7	51.4
HRD7		3.5	1.5	57.1
**Group mean**		**3.5b**	**1.55a**	**55.08b**

For each treatment, 15 replicate apple fruits were used, and the experiment was repeated twice. ** Mean values followed by different letters in the column indicate significant differences among treatments according to the pairwise Student’s *t*-test (*p* = 0.05).

**Table 2 plants-12-01398-t002:** Mean patulin production (μg/g), on PDA media (in vitro) and on artificially inoculated apple fruit (in vivo) of *Penicillium expansum* MDR and sensitive isolates in the absence or presence of the fungicide fludioxonil.

Isolate	Phenotype	In Vitro (μg/g) ^a^		In Vivo (μg/g) ^b^	
		Control Medium	Fludioxonil-Amended Medium	Untreated Fruit	Fludioxonil-Treated Fruit
HGS9	MDR	587.8	415.5	3.38	25.62
HF2		826.1	711.4	2.45	13.56
HF8		790.1	622.3	3.38	13.12
HF1		518.7	425.6	3.5	4.26
KRD6		616.4	504.6	9.5	8.6
KRD9		nd ^c^	nd	3.73	52.4
ARD7		576.8	489.5	11.18	4.6
ARD11		324.5	277.2	0.2	0.8
HG4		364.5	231.5	31.3	58.58
HGS12		351.7	347.5	32.5	19.86
HGS15		441.2	441	27.25	100.5
HGS16		252.7	205.7	18.49	130
KF3		396.2	212.7	5.54	40.3
AGS5		466.8	315.2	7.31	1.7
	**Group mean**	**465.2 ± 57.3a ^d^**	**371.4 ± 49.3a**	**11.4 ±** **3.0a ***	**25.5 ± 10.2a**
ZRD13	Sensitive	409.8	388.2	1.68	5.66
KF14		210.7	237.8	5.45	15.38
HF5		321.2	355.3	0.34	15.86
HRD7		372.4	328	13.35	30.2
	**Group mean**	**328.5 ± 43.2b**	**327.3 ± 32.2a**	**5.2 ± 2.9a ***	**16.7 ± 5.0** **a**

^a^ Mean values of yield of three replications. ^b^ Mean values of yield of two replications. ^c^ nd: not detected. ^d^ Mean values followed by different letters in the column indicate significant differences among MDR and sensitive isolates according to the pairwise Student’s *t*-test (*p* = 0.05). Mean values followed by asterisks in the rows indicate significant differences among control and fludioxonil treatments according to the pairwise Student’s *t*-test (*p* = 0.05).

## Data Availability

The data presented in this study are available in this article.

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
