# Peer review of "Aggressiveness and Patulin Production in Penicillium expansum Multidrug Resistant Strains with Different Expression Levels of MFS and ABC Transporters, in the Presence or Absence of Fludioxonil"

_plants, 2023, doi:10.3390/plants12061398_

Round 1

Reviewer 1 Report

The manuscript presents an interesting and original study aimed to assess the fitness of some MDR strains of Penicillium expansum versus wild sensitive to pesticides strains and to determine if MDR strains produce more mycotoxines (patulin). The results obtained are interesting as it appears that MDR strains are less aggressive but produce more patulin compared to wild strains, which is of health concern.

The study design is appropriate, and conclusions are supported by data. But some changes and revision are required.

Point 1: a recheck of entire document is needed as some typing and formatting errors occurred. Please have in mind lines 160-161 (italicize In vitro and in vivo), line 172, line 359 and other places in text.

Point 2: Recheck the formatting of reference list as multiple issues must be addressed!

Point 3: revise the phrase at lines 324-325. It is ambiguous.

Point 4: The phrase:

- “In our study, patulin production in most MDR isolates was significantly higher compared to that of isolates of wild-type both on artificial nutrient media (in vitro) and on apple tissue (in vivo).” at lines 279-281 should be revised, as a contradiction arise here with:

 No difference (P > 155 0.05) was observed in mean patulin concentration produced by the MDR and sensitive 156 isolates in the absence of fludioxonil in vivo, as the mean values were 11.4 and 5.20 μg/g, 157 respectively.” at lines 155-157.

Author Response

Response to Reviewer 1

The authors would like to thank the reviewer for the valuable suggestions and ideas that increased the quality of the manuscript. We really appreciate the fact the reviewer gave to us the opportunity to present our work in ''Plants''.

Below you can find a list of our responses to each point raised. We have incorporated these comments into the manuscript, and we thus hope that the manuscript can be accepted for final publication in ''Plants''.

Comment 1: a recheck of entire document is needed as some typing and formatting errors occurred. Please have in mind lines 160-161 (italicize in vitro and in vivo), line 172, line 359 and other places in text.

Response: The suggested change has been included into the manuscript.

Comment 2: Recheck the formatting of reference list as multiple issues must be addressed!

Response: The suggested change has been included in the manuscript.

Comment 3: revise the phrase at lines 324-325. It is ambiguous.

Response: The phrase has been revised on lines 324-326, to: “Such a contrast might be related with either a data inadequacy of significant genes expression in patulin biosynthesis pathway or with the impact of fludioxonil treatment on apple fruits”.

Comment 4: The phrase: “In our study, patulin production in most MDR isolates was significantly higher compared to that of isolates of wild-type both on artificial nutrient media (in vitro) and on apple tissue (in vivo).” at lines 279-281 should be revised, as a contradiction arise here with: “No difference (P > 0.05) was observed in mean patulin concentration produced by the MDR and sensitive isolates in the absence of fludioxonil in vivo, as the mean values were 11.4 and 5.20 μg/g, 157 respectively.” at lines 155-157.

Response: The phrase was revised according to the reviewer suggestion (see lines 279-282).

Reviewer 2 Report

Dear Authors

Please provide the e HPLC chards analysis as supplemental files.

Thanks 

Author Response

The authors would like to thank the reviewer for the valuable suggestions and ideas that increased the quality of the manuscript. We really appreciate the fact the reviewer gave to us the opportunity to present our work in ''Plants''.

Below you can find a list of our responses to each point raised. We have incorporated these comments into the manuscript, and we thus hope that the manuscript can be accepted for final publication in ''Plants''.

Comment: Please provide the HPLC charts analysis as supplemental files.

Response: The HPLC charts has been included in the manuscript as Supplementary Figure 1.

Reviewer 3 Report

The present Manuscript describes the aggressiveness and patulin production by strains of Penicillium expansum which is a pathogen of apples.

The Authors provide some interesting results on how the production of mycotoxin patulin and the aggressiveness of strains correlated with their earlier results published in

Samaras, Α.; Ntasiou, P.; Myresiotis, C.; Karaoglanidis, G.S. Multidrug resistance of Penicillium expansum to fungicides: whole 561 transcriptome analysis of MDR strains reveals overexpression of efflux transporter genes. Int. J. Food Microbiol. 2020, 335, 108896. 562 https://doi.org/10.1016/j.ijfoodmicro.2020.108896.

The amount of data obtained can be published. Unfortunately, the present Manuscript doesn’t provide the complete picture of the events, lacks essential details of experiments and requires complete rewriting before being considered further.

For example, description of Table S2 “Oligonucleotides used in this study to induct of patulin biosynthesis related genes in vitro and in vivo samples with qRT-PCR” doesn’t correspond to the experiments done.

Table S1 needs more detailed description in the figure legend which solvents were present in the phases described.

Figure 1 requires scale bar for the size of the fruits used, then the lesion diameters cannot be determined from the pictures since the lesion spots overlap. Please, provide the other picture and indicate why the given picture does not correspond to the methods

“The fruit were wound-inoculated at the equator. Three 368 wounds per fruit were made using a 2-mm diameter nail head into 3 mm in depth and on 369 each wound 20 μl of a conidial suspension containing 1×105 conidia per ml were placed 370 with a pipette.”

Statistical treatments are not shown for any of the figures and tables where they should be given.

The term overexpression means typically using special artificial laboratory tools to cause overexpression.

“The MDR phenotype 357 of the resistant isolates was associated with overexpression of MFS and ABC transporter 358 genes at variable levels.”

Overexpression of the genes needs to be confirmed by mutants or overexpressing isolates, otherwise the correlation is not sufficient for the statements.

The isolates should be well described initially as well as the opportunities to get them for reproducing the experiments.

Methods. Please, indicate the number of fruits per a box and correct the sentence. “The fruit were placed in plastic boxes (23×31×10 cm 373 [length×width×height]).”

The text requires logical re-structuring before being considered further on.

The language needs to be improved.

Author Response

Response to Reviewer 3

The authors would like to thank the reviewer for the valuable suggestions and ideas that increased the quality of the manuscript. We really appreciate the fact the reviewers gave us the opportunity to present our work in the ''Plants'' Journal.

Below you can find a list of our response to each point raised. We have incorporated these comments into the manuscript, and we thus hope that the manuscript can be accepted for final publication in ''Plants''.

Comment 1: Description of Table S2 “Oligonucleotides used in this study to induct of patulin biosynthesis related genes in vitro and in vivo samples with qRT-PCR” doesn’t correspond to the experiments done.

Response: The Table S2 legend was revised (see new legend). In this Supplementary Table we provided information on the oligonucleotides sequence used to quantify the expression levels of patulin-biosynthesis genes.

Comment 2: Table S1 needs more detailed description in the figure legend which solvents were present in the phases described.

Response: The solvents of mobile phases have been added as footnote in Supplementary Table 2, according to the reviewer suggestion.

 Comment 3: Figure 1 requires scale bar for the size of the fruits used, then the lesion diameters cannot be determined from the pictures since the lesion spots overlap. Please, provide the other picture and indicate why the given picture does not correspond to the methods. “The fruit were wound-inoculated at the equator. Three wounds per fruit were made using a 2-mm diameter nail head into 3 mm in depth and on each wound 20 μl of a conidial suspension containing 1×105 conidia per ml were placed with a pipette.”

Response: Indeed, overlapping exists in one of the 4 figure panels (Fig 1B). But we preferred to include this photο in order to show the higher pathogenicity of the wild type strains. In addition, the reviewer has right suggesting that there was an inconsistency between the Figure and the description in methods. Actually, the artificial inoculation was made above or below the fruit equator. Therefore, the respective description in the materials and methods section was changed (see lines  371-372).  

Comment 4: Statistical treatments are not shown for any of the figures and tables where they should be given.

Response: Information about the statistical analysis on our data, have been included in all the Figure legends or Table footnotes, where it was required (see lines 137, 167, 187, 207 and 225). If something more is required, please specify.

Comment 6: “The MDR phenotype of the resistant isolates was associated with overexpression of MFS and ABC transporter genes at variable levels.” Overexpression of the genes needs to be confirmed by mutants or overexpressing isolates, otherwise the correlation is not sufficient for the statements.

Response: The reviewer has right suggesting that overexpression of the genes needs to be confirmed in mutant isolates. Indeed, in general, functional analysis experiments with mutants give the absolute correlation between a treatment and a biological pattern. Such mutants are lacking in our study and would be an excellent addition in our paper strengthening our data.   However, in our study was used a quite high number of isolates with increased expression rates of ABC and MFS transporters confirmed by RNA seq analysis coupled with quantitative PCR data (all these data have been included in our recent paper, cited in our article -Samaras et al 2020 IJFM, that served as the basis for the work that we present in the current paper). By using a quite high isolate number we provide an indirect indication that this pattern is not an artifact.

Comment 7: The isolates should be well described initially as well as the opportunities to get them for reproducing the experiments.

Response: As mentioned in the Materials and Methods section, the isolates used in the study had been characterized by RNA seq and qPCR analysis for the requirements of a previous study in which the phenotype had been described in detail (lines 357-360).

Comment 8: Methods. Please, indicate the number of fruits per a box and correct the sentence. “The fruit were placed in plastic boxes (23×31×10 cm [length×width×height]).”

Response: The number of the fruits per box has been included in the manuscript (see line 375).

Comment 9: The language needs to be improved.

Response: The manuscript has been edited by a professor of English language and literature.

Round 2

Reviewer 2 Report

Dear Author

The manuscript looks good after taking the reviewer's comments. I accept it to be published in the present format.

Author Response

The authors would like to thank the reviewer for the valuable suggestions and his/her final recommendation for acceptance of the paper. We really appreciate the fact the reviewer gave to us the opportunity to present our work in ''Plants''.

Reviewer 3 Report

The present Reviewer is glad to encounter good and promising progress with the revisions. The Reviewer leaves the fate of the MS to the Editor. I am sorry for not being more positive this time. The results are good and could be published.

The present suggestions with questions are:

1) to change the title to “Aggressiveness and patulin production in Penicillium expansum multidrug resistant strains with different expression levels of MFS and ABC transporters, in the presence or absence of fludioxonil” or a similar title since the overexpression was not demonstrated in the presented results of the study;

2) to add scale bar in nautical miles or microns or in any units of linear measurements for figure 1;

3) to add the number of replicates if any for table 1; the same is for table 2;

4) to add the error bars for figures with gene expression levels that is for all the figures apart from figure 1.

5) Please, use the uniform style for references.

Table S2 is good though extra reference genes could be and preferably should be checked.

6) Figure S1 for HGS16 and fruits doesn’t agree with results from table 2 for the isolate.

7) Penicillium expansum should be in italics (e.g. legend for table 2).

8) Figure 1 is bad for the publication to reveal distinct spots.

Author Response

The authors would like to thank the reviewer for the valuable suggestions and ideas that increased the quality of the manuscript. We really appreciate the fact the reviewer gave to us the opportunity to present our work in ''Plants''.

Below you can find a list of our responses to each point raised. We have incorporated these comments into the manuscript, and we thus hope that the manuscript can be accepted for final publication in ''Plants''.

Comment 1: To change the title to “Aggressiveness and patulin production in Penicillium expansum multidrug resistant strains with different expression levels of MFS and ABC transporters, in the presence or absence of fludioxonil” or a similar title since the overexpression was not demonstrated in the presented results of the study.

Response: The suggested change has been included in the manuscript. (see the title)

Comment 2: To add scale bar in nautical miles or microns or in any units of linear measurements for figure 1.

Response: The suggested change has been included in the manuscript. The scale bar has been added (see Figure 1).

Comment 3: To add the number of replicates if any for table 1; the same is for table 2;

Response: The suggested changes have been included in the manuscript. The number of replicates has been added as footnote in both Tables.

Comment 4: To add the error bars for figures with gene expression levels that is for all the figures apart from figure 1.

Response: The suggested change has been included in the manuscript. The error bars have been added in Fig. 2, Fig. 3 and Fig.4.

Comment 5: Please, use the uniform style for references.

Response: We recheck the format of the references. The reference list was prepared according to journal guidelines.

Comment 6: Table S2 is good though extra reference genes could be and preferably should be checked.

Response: The reviewer has right suggesting that, in general, more reference genes preferably should be checked. However, we followed the standard practice of testing 1 reference gene.  We thought that it was not necessary to add more genes, since b-tubulin is a well-studied reference gene for Penicillium species. Numbers of studies mentioned that this gene remains stable across various treatments and the values are accurate for normalization.

Comment 7: Figure S1 for HGS16 and fruits doesn’t agree with results from table 2 for the isolate.

Response: Thank you for the observation. The correct image has been included in the Figure.

Comment 8: Figure 1 is bad for the publication to reveal distinct spots.

Response: Τhe image quality is according to the guidelines, and the figure has the appropriate for the journal dpi (dots per inch). In addition, this picture showed the important differences between resistant and sensitive strains of P. expansum in the absence and in the presence of fludioxonil.
